# "*I see salt everywhere*": A qualitative examination of the utility of arts-based participatory workshops to study noncommunicable diseases in Tanzania and Malawi

Maria Bissett[1,2], Cindy M. Gray[1,2]*, Sharifa Abdulla[3], Christopher Bunn[1,2,4], Amelia C. Crampin[2,4,5], Angel Dillip[6], Jason M. R. Gill[7], Heri C. Kaare[8], Sharon Kalima[9], Elson Kambalu[10], John Lwanda[1], Herbert F. Makoye[8], Otiyela Mtema[11], Mia Perry[12], Zoë Strachan[13], Helen Todd[9], Sally M. Mtenga[6]

1 School of Social and Political Sciences, University of Glasgow, Glasgow, United Kingdom, 2 School of Health and Wellbeing, University of Glasgow, Glasgow, United Kingdom, 3 Fine and Performing Arts Department, University of Malawi, Zomba, Malawi, 4 Malawi Epidemiology and Intervention Research Unit, Lilongwe, Malawi, 5 London School of Hygiene and Tropical Medicine, London, United Kingdom, 6 Health Systems, Impact Evaluation and Policy, Ifakara Health Institute, Dar es Salaam, United Republic of Tanzania, 7 School of Cardiovascular & Metabolic Health, University of Glasgow, Glasgow, United Kingdom, 8 Taasisi ya Sanaa na Utamaduni Bagamoyo (TaSUBa), Bagamoyo, United Republic of Tanzania, 9 Art and Global Health Centre Africa, Zomba, Malawi, 10 Art House Africa, Lilongwe, Malawi, 11 Zaluso Arts, Lilongwe, Malawi, 12 School of Education, University of Glasgow, Glasgow, United Kingdom, 13 School of Critical Studies, University of Glasgow, Glasgow, United Kingdom

* Cindy.gray@glasgow.ac.uk

**Data Availability Statement:** All qualitative data relating to the findings are included in the paper in the form of quotations and figures. The ethical approvals from the Ethical Committee at the

## Abstract

The burden of noncommunicable diseases (NCDs) including hypertension, diabetes, and cancer, is rising in Sub-Saharan African countries like Tanzania and Malawi. This increase reflects complex interactions between diverse social, environmental, biological, and political factors. To intervene successfully, new approaches are therefore needed to understand how local knowledges and attitudes towards common NCDs influence health behaviours. This study compares the utility of using a novel arts-based participatory method and more traditional focus groups to generate new understandings of local knowledges, attitudes, and behaviours towards NCDs and their risk factors. Single-gender arts-based participatory workshops and focus group discussions were conducted with local communities in Tanzania and Malawi. Thematic analysis compared workshop and focus group transcripts for depth of content and researcher-participant hierarchies. In addition, semiotic analysis examined the contribution of photographs of workshop activities to understanding participants' experiences and beliefs about NCD risk factors. The arts-based participatory workshops produced in-depth, vivid, emotive narratives of participants' beliefs about NCDs and their impact (e.g., "*. . . it spreads all over your body and kills you—snake's poison is similar to diabetes poison*"), while the focus groups provided more basic accounts (e.g., "*diabetes is a fast killer*"). The workshops also empowered participants to navigate activities with autonomy, revealing their almost overwhelmingly negative beliefs about NCDs. However, enabling participants

National Institute for Medical Research (NIMR) in Tanzania, the National Committee on Research in the Social Sciences and Humanities in Malawi and the University of Glasgow College of Social Science Ethics Committee do not give us license to make the dataset publicly available. However, data may be made available on request to the corresponding author and/or the University of Glasgow College of Social Science Ethics Officer, socsci-ethics@glasgow.ac.uk

**Funding:** This study was part of a larger project exploring interdisciplinary approaches to NCD prevention in Tanzania and Malawi (Culture and Bodies) jointly funded by the UK's Medical Research Council and Arts and Humanities Research Council- reference number MC_PC_MR_R024448, 21. The funders had no role in study design, data collection and analysis, decision to publish, or preparation of the manuscript.

**Competing interests:** The authors have declared that no competing interests exist.

to direct the focus of workshop activities led to challenges, including the perpetuation of stigma (e.g., comparing smells associated with diabetes symptoms with sewage). Semiotic analysis of workshop photographs provided little additional insight beyond that gained from the transcripts. Arts-based participatory workshops are promising as a novel method to inform development of culturally relevant approaches to NCD prevention in Tanzania and Malawi. Future research should incorporate more structured opportunities for participant reflection during the workshops to minimise harm from any emerging stigma.

## Introduction

Communicable diseases such as malaria and HIV/AIDS have traditionally been the primary focus of public health initiatives in Sub-Saharan Africa [1]. However, the additional burden of noncommunicable diseases (NCDs) is now of growing concern [2, 3]. Between 1990 and 2017, NCDs accounted for a 67% increase in disability-adjusted life-years in Sub-Saharan Africa [4] and current estimates predict that NCD mortality will outstrip that from communicable, maternal and perinatal diseases by 2030 [5].

Tanzania and Malawi, like other Sub-Saharan Africa countries, are experiencing major challenges in the face of this additional burden of disease [6, 7]. In 2019, NCDs were estimated to account for 34% of deaths in Tanzania and 40% in Malawi [8]. Recent studies suggest that prevalence of hypertension is high (28%) in parts of Tanzania and uncontrolled among 95% of patients [9], and up to 6.8% of adults have diabetes [10]. Similarly, both hypertension (33%) and diabetes (5.6%) are highly prevalent and underdiagnosed in Malawi [7, 11].

The drivers of many NCDs are complex and caused by interactions between social, environmental, biological, and political factors [2]. Evidence indicates there may be a predisposition to poor metabolic health associated with historic and persistent undernutrition and communicable disease in low-middle income countries like Tanzania and Malawi [2, 3]. In addition, there is an increasingly obesogenic environment due to the commercial influx of calorific, low nutrient foods, alcohol, and tobacco, while structural barriers associated with rising urbanisation in the region may further restrict access to healthy food and spaces to exercise [2]. Finally, different socio-cultural norms, such as a large body size being associated with wealth and health [3] and traditional food preferences [12, 13] mean that the lifestyle interventions used to address NCDs in high-income countries may not be translatable to the Sub-Saharan African context [1]. Culturally relevant approaches are needed to understand how local beliefs and culture influence health behaviours in Tanzania and Malawi to inform effective NCD prevention initiatives.

Historically, health research in Sub-Saharan Africa has prioritised a Western scientific approach to understand the lived realities of local communities [14, 15]. This approach has generated mistrust of Western researchers among local people [14, 16]. New methodologies should therefore seek to foreground the cultural significance of local people's understandings of health and, where possible, repair any damaged trust between communities and researchers [17, 18].

Community-based participatory research (CBPR) has emerged to promote equitable collaborations between health researchers and study participants [18]. Genuine partnerships between researchers and community members, where each partner benefits from mutual knowledge generation, are key to CBPR, which has been used successfully with marginalised groups in high-income contexts [18]. The use of CBPR in Sub-Saharan Africa can be traced

back to earlier participatory approaches in the 1970s, including action research and Theatre for Development [19–25]. Although CBPR approaches can bring new learning opportunities and rich perspectives for those involved, there are still many practical challenges with ensuring that participation is truly equitable [18, 23, 26].

Arts-based methods are also becoming increasingly used within health research. They incorporate a range of artistic practices including performance, visual arts and written narratives, and can be used at any stage of the research process [27] to enable the communication of rich, affective narratives that are difficult to express through words alone [28]. Arts-based methods have been widely used in communicable disease research in Sub-Saharan Africa, particularly HIV/AIDS, but remain largely underutilised in NCD research [17]. However, some researchers have criticised the use of arts-based research in Sub-Saharan Africa for unintentionally reinforcing researcher-participant hierarchies. Examples include failing to actively engage performers in creating the narratives [29] and communicating health messages that prioritise western biomedical knowledge over the lived experiences of local communities [17].

Integrating arts-based methods and CBPR provides a promising way of overcoming these challenges. Some researchers have demonstrated the utility of participant-centred arts-based approaches in narrowing the distance between the researcher and the researched [30] and liberating 'performers' to express themselves about the issues they find most meaningful and important [27, 28, 31]. Building on this evidence, the current study sought to examine the utility of arts-based participatory workshops in exploring local knowledges, attitudes, and behaviours towards common NCDs and their risk factors in Tanzania and Malawi. The aim was to compare the workshops to traditional focus group discussions in relation to the depth of content generated and researcher-participant hierarchies. A secondary aim was to examine the extent to which visual data (photographs) from the workshops contributed to understanding local experiences and beliefs around NCD risk factors.

## Methods

### Ethics statement

Ethical approvals were obtained from the Ethical Committee at the National Institute for Medical Research (NIMR) in Tanzania (NIMR/HQ/R.8a/Vol.IX/2959), the National Committee on Research in the Social Sciences and Humanities in Malawi (Ref. NCST/RTT/2/6), and the University of Glasgow College of Social Science Ethics Committee (Ref. 400170232). Participants were provided with information sheets containing full details of the project in their local language. The facilitator explained that participation was fully voluntary and summarised the information sheet before prompting participants to ask questions if they required further clarification or information. All participants provided written informed consent. The individuals pictured in Fig 2 have also provided written informed consent (as outlined in the media consent form) to publish their image alongside the manuscript.

### Study design

The current study is part of a larger MRC/AHRC-funded project exploring interdisciplinary approaches to NCD prevention in Tanzania and Malawi (Culture and Bodies, MC_PC_MR_R024448, 21). Culture and Bodies was a collaboration between researchers from Ifakara Health Institute in Tanzania, the Malawi Epidemiology and Intervention Research Unit (MEIRU), the University of Malawi, the University of Glasgow; and local participatory arts practitioners from Bagamoyo Arts and Cultural Institute (Taasisi ya Sanaa na Utamaduni Bagamoyo, TaSUBa) in Tanzania and Art and Global Health Centre Africa (ArtGlo) and Art House Africa in Malawi.

The current study involved three main components. First, community engagement activities were undertaken to allow the researchers to develop relationships with the local communities and generate initial insight into local NCD knowledge and community arts. Then focus group discussions and arts-based participatory workshops were used to explore local attitudes to NCDs and risk factors. This paper reports the qualitative findings from the focus group discussions and arts-based participatory workshops.

## Setting and participants

The study was conducted in Bagamoyo, Tanzania and Area 25, Lilongwe, Malawi. Bagamoyo is a coastal town in Tanzania with a population of 89,000 in 2008 [32]. Area 25 is a high-density area of the Malawian capital, Lilongwe, with a population of 64,650 in 2008 [33]. Both communities have a history of involvement in previous health research, and therefore may be more familiar with health issues than other similar communities.

Four focus group discussions (N = 7–10) and four arts-based participatory workshops (N = 7–10) were conducted in single-gender groups to minimise the impact of cultural gender norms on participants' contributions. Local residents were recruited using a convenience sampling approach with the help of community leaders. All were aged over 18 years and had lived in the community for at least 12 months. Participants were initially invited to take part in the focus group discussions and then invited to also attend the arts-based participatory workshops. Additional participants were recruited as required to make up numbers for the workshops. Some participants already knew each other; however, most met for the first time during the research.

## Data collection and analysis

The arts-based participatory workshops and focus group discussions aimed to explore people's knowledges, attitudes, and behaviours towards NCDs and their risk factors including symptoms, and prevention. In Tanzania, the workshops (November/December 2018) were facilitated by SMM with support from AD, CMG and HCK, and the focus groups (February/March 2019) were conducted by SMM and AD. In Malawi, the workshops (February 2019) were co-facilitated by SK and OM, and the focus groups (February 2019) were co-conducted by OM and HN. A topic guide was used to steer the focus group discussions, which were designed to last around 1.5–2 hours, around the issues of interest, including a range of common NCDs.

In the workshops, participants were asked to select one NCD to focus on. In Tanzania, both groups chose to explore diabetes; in Malawi, the men selected hypertension and the women chose cancer. Half a day was allocated to the workshops, as the enjoyable nature of the creative activities lessened the burden on participants. As shown in Fig 1, each workshop consisted of three activities (*Verbalised senses, Embodied images, and Performance)*, each followed by reflective discussion. The activities were designed to build on each other, with the first two providing a scaffold to support the development of the performance activity. Embodied images and performances were created in small groups and interpreted by other participants. For the final performance, participants could use an arts-based approach of their choice. Theatre was selected in three of four workshops, while men in Tanzania chose poetry.

The focus groups and workshops were audio-recorded, transcribed, and translated from the local languages (Kiswahili and Chichewa) to English by local fieldworkers. Fieldnotes were also taken during each method. Photographs of the embodied images were taken in the Tanzanian women's workshops and both Malawi workshops.

The transcripts were analysed using a thematic framework approach [34] and NVivo12 software. One focus group and one workshop transcript were independently coded by MB,

## Activity 1: Verbalised Senses

Participants were asked to reflect on what the NCD of their choice:

NCD looks like...

NCD sounds like...

NCD smells like...

NCD tastes like...

NCD feels like...

## Activity 2: Embodied Images

In pairs, participants were asked to use their bodies to create a still image exploring lifestyle risk factors of NCDs.

## Activity 3: Performance

In larger groups, participants were asked to create a performance about the causes and consequences of the lifestyle risk factor in the embodied image

Create a series of actions that led to the image

Create a series of actions that follow the image

Perform the whole action sequence to the rest of the group.

**Fig 1. Arts-based participatory workshop activities.** Description of the arts-based participatory workshop verbalised senses, embodied images, and performance activities.

CMG and SMM, who then met to agree nine broad themes for analysis, six of which are reported here: Beliefs about NCDs and risk factors; Impact of NCDs; Experiences of NCDs and risk factors; Causes of NCDs and risk factors; Barriers to NCD prevention; and NCD symptoms. A full description of all nine themes is provided in S1 Table. The broad themes were then applied to all transcripts by MB and checked by CMG. MB generated relevant subthemes and applied them to Excel matrices. Data were then sorted according to method and each sub-theme summarised with focus on the similarities and differences between the arts-based participatory workshops and focus group discussions. The comparisons were aligned with the main aims of the research; specifically, depth of content and researcher-participant hierarchy.

Semiotic analysis was used to analyse eleven photographs of embodied images from the workshops. Semiotics conceptualise images as a series of signs (e.g., body language) to understand how meaning is created [35, 36]. MB applied semiotic analysis to explore the

characterisation and the emotions portrayed within the images as follows: first, literal descriptions of what the characters were doing in the images were written without assigning any further meaning (e.g., the character holds a bottle to their lips vs. the character is drinking alcohol). The images were then broken down further into visual sign categories (i.e., physical appearances of actors and body language/facial expressions) summarised in an Excel matrix. As cultural knowledge is key to assigning higher levels of meaning, the matrix also referenced the relevant participant discussions within transcripts [35].

The first author (MB) is a Western researcher who was not present at data collection. Analysis was, therefore, approached critically with care taken to recognise and correct any biases and misinterpretations. It was closely supported by Culture and Bodies investigators, including SMM, a Tanzanian national.

## Results

The analysis compared arts-based participatory workshops to traditional focus group discussions in relation to i) depth of content generated; and ii) researcher-participant hierarchies during each method. Relevant findings to i) and ii) are therefore summarised and presented in turn below.

### i) Depth of content

The workshops stimulated participants to use vivid and emotive imagery, which often generated more insight into their own deeply held beliefs about the chosen NCDs in comparison to the focus groups. This was especially apparent in the verbalised senses activities: for example, participants equated the chosen NCD to a *"bomb"*, *"bullet"* and *"plane crash"*. This language suggests sudden, catastrophic impacts of the diseases as depicted in the following workshop extract from Tanzania where men used powerful similes to portray amputation as a complication of diabetes:

> "*Diabetes looks like someone coming to you with a bomb and want to kill you, because it is like a bomb that can take some parts of your body like legs, limbs. Like your enemy coming with a bomb*" (Men's Workshop Tanzania)

By comparing diabetes to a bomb, the participant conveys a sense of destruction. This analogy strongly evokes feelings of fear, worry and pain, and therefore carries more weight than simply saying that limb amputation is a negative impact of diabetes. Others communicated beliefs of the chosen NCDs being like predatory animals, including a *"hyena"*, *"lion"* and *"snake"*. Animals play important roles in African cultures, often described in symbolism, folklore, and spirituality [37–39]. Although the associated traits may vary between cultures, the hyena is usually depicted as an antagonistic symbol, described as sly and unintelligent [37, 39]. The lion is frequently considered a mark of power and strength [39], and, in some circumstances, danger [40]. When reflecting on these ideas, one Tanzanian man explained that their belief stemmed from the perceived impact of the disease:

> "*I associate a snake with diabetes because the poison of the snake goes through your body for just few minutes, same as diabetes once you have it, it spreads all over your body and kills you —snake's poison is similar to diabetes poison.*" (Men's Workshop, Tanzania)

This extract evokes a sense of powerlessness to diabetes, in which fatal consequences are unavoidable. In Sub-Saharan Africa, communities face several barriers to controlling diabetes, including regular access to medications [13]. Therefore, the powerful association of diabetes

with a snake's poison could reveal both beliefs about the severity of the disease and the lived experience of barriers to controlling it. In contrast, the focus groups provided more basic 'bare-bones' accounts of the impacts of NCDs:

> "*Diabetes it's a fast killer once its high you can die right away, Cancer we have no cure in Malawi.*" (Women's Focus Group, Malawi)

In some cases, while the workshops provided rich analogies, participants could have been supported to reflect on emerging beliefs more fully. During the verbalised senses activity in Tanzania, one male participant remarked; *"When I see diabetes, I see salt everywhere"*. This statement evokes a vivid mental image that could be interpreted as salt being perceived as causing diabetes. Salt is a well-documented risk factor for hypertension, but not for diabetes [41]. However, hypertension and diabetes present together in a high proportion of patients [42–44]. Therefore, the analogy could also signify familiarity with the healthy eating advice provided to patients with diabetes in Sub-Saharan Africa, which typically includes salt-reduction to prevent or control comorbid hypertension [45, 46]. However, on this occasion, the participant was not asked to elaborate, and his intended meaning remained unclear.

Where opportunity was provided for reflection, the workshop activities helped participants to consider alternative causes of NCD risk behaviours. They produced emotive and empathetic insight compared to the focus groups. For example, following a performance about alcohol leading to high blood pressure in Malawi, participants had different opinions on the *"drunkard"* character. Some argued that the character's life choices made him a poor and unhelpful husband. However, when prompted to consider different scenarios about why somebody would drink, others imagined a backstory for the character and appeared more compassionate towards him:

> "*Sometimes there are problems that leads him into drinking, maybe the things he faces at the household*" (Men's Workshop, Malawi)

While the workshops provided more emotive insight into local experiences of NCDs, the focus groups provided important narratives on topics not discussed in the workshops. These included reflections on how circumstances outwith personal control could act as barriers to prevention. For example, when asked what type of cooking oil was used locally, one Tanzanian man complained about the prohibitive cost of healthier options:

> "*P2: We use korie [refined vegetable] cooking oil, but I think sunflower oil is the best, but we cannot afford. It is expensive. A small bottle at 4,000 Tsh. Where do I get that money?*" (Men's Focus Group, Tanzania)

This extract demonstrates that knowledge about NCD risk factors alone is insufficient to change behaviour, and the use of a rhetorical question gives a sense of the frustration and lack of empowerment experienced by the community to do the 'right thing'. Thus, whilst the workshop activities appeared to facilitate participants' exploration of personal and wider social factors that contribute to NCD risk, the focus groups tended to focus more on macro level structural factors, such as unaffordability of healthy lifestyle practices.

### ii) Researcher-participant hierarchies

Focus group participants covered a broad range of topics. For example, when considering the causes of NCDs, some traditional widely held beliefs around their supernatural and fatalistic

nature emerged. However, as the following extract indicates, these beliefs were often elicited only after direct prompts from the researcher:

*I: I want you to tell me what people say about these diseases in your communities?*

*P6: People say they are uncurable disease.*

*P5: For people who have not gone to the hospital with their child who has brain disorder they say the child was bewitched.*

*P7: Once you have cancer in our communities, they say the patient should just wait for his/her death.* (Women's Focus Group, Malawi)

In contrast, the workshop's verbalised senses activity was particularly valuable when allowing participants to describe their chosen NCD with autonomy. Most participants used this activity to highlight negative beliefs, indicating that they themselves recognise the increasing challenge of NCDs in their communities. For example, Malawian men associated hypertension with things that were dangerous (*"axe"*, *"fire"*) and startling (*"thunder"*, *"fireworks"*), leading one participant to express his surprise at the extent of the negativity that had emerged from the group:

"*It has really shown me that BP* [high blood pressure] *is a very dangerous disease, by the [. . .] way everyone has described BP, no one has said a good thing about BP like, it tastes like Ice cream, or it feels like your Darling, completely nothing good"*. (Men's Workshop, Malawi)

Encouraging autonomy during the workshops also brought unexpected challenges, particularly in relation to the focus of the workshop activities. For example, the Malawian women selected cancer as their NCD of choice, but then explored a range of cancers (skin, cervical and prostate) during the embodied images and performance activities, this restricted the opportunity for the research team to gain depth of insight into local knowledges, attitudes, and behaviours in relation to a single type of cancer. Another unintended consequence of allowing participants to direct the focus of the workshop activities was that their imaginative reflections appeared to perpetuate stigma to a greater degree than the focus group discussions. In Tanzania, diabetic symptoms, such as frequent urination and slow-healing wounds were raised in both men's and women's focus groups. For example, in the women's focus group in Tanzania it was simply stated that wounds *"smell a lot if not treated"*. However, in the workshops, the verbalised senses activity stimulated participants to provide a complex narrative around the negative social norms associated with people with diabetes:

"*I said that diabetes smell like 'Karo la city' (sewage tank in the city) because usually the diabetic patients have wounds, and those wounds stays for a long time and smell badly. Sometimes their relatives segregate these patients because of bad smell. You know it is also difficult to eat near the sewage tank"* (Women's Workshop, Tanzania)

The comparison to a sewage tank is a rich analogy that fully portrays local perceptions of the unpleasant smells associated with people with diabetes. They do not merely smell 'bad' but appear to be intolerable to the point of disrupting social interactions. Similarly, the Tanzanian men's reflections on the verbalised senses activity drew parallels between diabetes and sewage (*"a pit with a strong smell"*) which was later reinforced within their final performance (poem):

"*Once you are diabetic you will disturb your children*

*Bringing tins every time for you to pee in*

*The smell will disturb them*

*Because you are their father, they will not run away from you"*

(Men's Workshop, Tanzania)

Therefore, although the workshop activities provided the researchers with a deeper understanding of local experiences of diabetes, they could also perpetuate stigma by providing validation of participants' negative views.

### iii) Using visual data to explore local understandings of NCD risk factors

A secondary aim of the study was to examine the extent to which photographs of the workshop embodied images activity could contribute to developing researchers' understanding of local experiences and beliefs around NCD risk factors. The semiotic analysis found that characterisation (physical appearances) had some utility. For example, as Fig 2 shows, the Tanzania women's workshop chose a participant with a larger body size to portray a physically inactive individual with diabetes, thus also suggesting the recognition of overweight/obesity as a risk factor for diabetes locally. However, the analysis of emotion (through body language and facial expression) was often inconclusive or did not contribute any further understanding beyond what was available from workshop transcripts. Indeed, interpretation of the visual data was often only possible with the addition of important cultural insights from the transcripts.

## Discussion

In comparison to focus groups, arts-based participatory workshops generated more in-depth insight into local knowledges, attitudes, and behaviours around NCDs and their risk factors. In relation to depth of content, the workshop activities evoked strong emotive portrayals of participants' fear, worry and powerlessness in relation to NCDs. Reflective discussions encouraged workshop participants to further explore and examine their own and other group members' perspectives of NCDs and NCD risk factors. However, on occasion, reflective discussions needed further development to clarify ambiguity around the intended meanings of participants' symbolic representations. Focus group discussions, on the other hand, provided insight into more tangible concepts and the structural causes of NCDs. In addition, workshops clearly reduced researcher-participant hierarchies and supported participants' autonomy around their chosen NCD, particularly in relation to their negative beliefs. However, this autonomy may have led to greater stigmatisation of the NCD than was apparent in the focus group discussions by providing a platform for participants to gain validation of their negative views. Finally, although photographs of workshop activities provided some additional understanding of local experiences and beliefs of NCD risk factors, the contribution of these visual data was enhanced by the cultural insights present in the workshop transcripts.

It has been argued that arts-based methods enable participants to delve into the imaginary, go beyond literal depictions of illness and provide new in-depth perspectives compared to traditional forms of research [28, 47, 48]. This was particularly true in the current study in relation to the emergence of fears and negative beliefs about NCDs, and the description of symptoms (e.g., diabetic wounds) during the verbalised senses activities. Our findings align with previous arts-based research: for example, Horne [49] noted that metaphors generated through body mapping and storytelling enabled HIV-positive women in South Africa to communicate deep pain about the lived realities of their condition, as well as confront their personal experience of the disease. Similarly, Abdulla [40] emphasised how 'play' and 'make

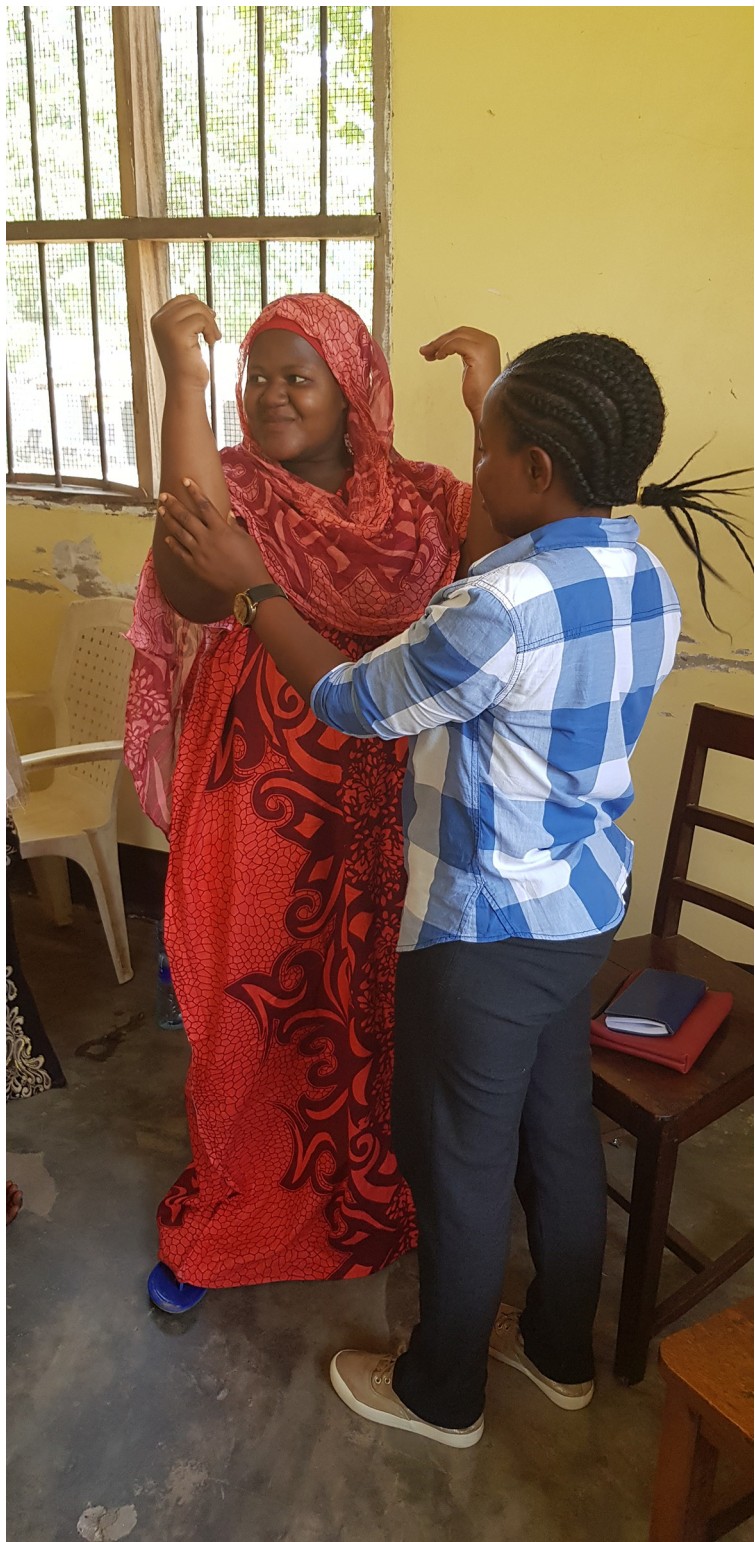

**Fig 2. Embodied image from the women's arts-based participatory workshop in Tanzania.** One woman supporting the arms of another woman to depict a lack of exercise as a cause of diabetes. The individuals pictured have provided written informed consent (as outlined in the media consent form) to publish their image alongside the manuscript.

believe' during community arts workshops in rural Malawi contributed to creating safe spaces between reality and the imaginary where participants could speak openly about their perspectives and experiences of HIV/AIDs without fear of real-life consequences.

It has been argued that where community members are directly involved in the creation of arts-based activities, they may be better able to explore and navigate the new knowledge produced [24]. In the current study, using verbalised senses and embodied images as a scaffold for participants to create their performances appeared to empower them to actively engage in the workshop in a way that was consistent with their personal interests and concerns. In addition, opportunities for reflection throughout the workshops provided participants with space to consider the new knowledges emerging from the activities (e.g., the overwhelmingly negative response to hypertension in the Malawian men's workshop). Our approach echoes a previous study where a creative, reflective participatory workshop conducted as part of an exploration of community resilience in the Netherlands encouraged participants to understand and share each other's feelings and discover common concerns. These findings led the authors to conclude that the creative empathy which emerged during the workshop supported in-depth communication of participants' attitudes and feelings in a way that moved beyond "cognitive ways of knowing" (pg.6; [28]).

Some researchers have suggested that narratives stimulated by arts-based methods can further marginalise vulnerable individuals [27, 50]. During the workshops, the combination of the arts-based activities providing participants with a 'safe space' to demonstrate negative beliefs about NCDs and the reflective discussion encouraging creative empathy may have unintentionally reinforced NCD-related stigma. However, arts-based methods have been shown to be helpful in challenging health-related stigma in Sub-Saharan Africa [17, 51]. For example, one photovoice study in South Africa encouraged secondary school students to use their own portrayals of HIV/AIDS-related stigmatisation to reflect on misunderstandings and negative attitudes towards people affected by HIV/AIDS [50]. Future workshops should include additional reflective activities to encourage participants to consider how their narratives could contribute to stigma and explore how they might support people who experience it.

Another challenge associated with participant autonomy during one of the workshops emerged in relation to cancer being the chosen NCD. This led to somewhat superficial considerations of different types of cancer rather than in-depth consideration of one. Howard [52] reported similar difficulties when using interactive theatre as a vehicle to generate information about eating habits and body image among US women. Spectators redirected the intended focus of the performance from broader social issues to individual situations and events that reflected their own personal experiences rather the collective viewpoint.

The current study was facilitated by close collaboration between researchers, arts-practitioners and community members from Tanzania, Malawi, and Scotland. Such cross-cultural interdisciplinarity means grappling with diverse epistemologies on what constitutes truth and different definitions of the topic being researched [53, 54]. Navigating what and how knowledge is shared also requires critical awareness of where participant contributions and researcher goals conflict [54]. Some of the challenges outlined above, regarding the workshop focus and need for different forms of reflection, could also reflect the complexity of navigating the interests of multiple disciplines and partners and expectations of what the collected data should look like. These examples serve as a reminder of the importance of transparency, reflexivity and dialogue between different partners to explore these challenges [54, 55].

Finally, the contribution of the photographs of the embodied images activity to our understandings of local experiences and beliefs about NCD risk factors was limited. When exploring emotions, it was sometimes difficult to ascertain to what extent the emotions conveyed in the photographs were intended as part of the image or were spontaneous facial expressions that

emerged from taking part in the activity. This limitation could be overcome by using the principles of semiotic analysis (e.g., physical appearances of actors and body language/facial expressions) during future workshops to guide participants' reflective interpretations of images and performances. This approach would be beneficial in further empowering participants by supporting their involvement in the interpretation of the data generated and thus the production of culturally relevant insight [27, 56].

## Strengths and limitations

A major strength of this study is the involvement of different teams of local researchers, arts practitioners, and community members in two Sub-Saharan Africa countries. This not only provides important insights into how arts-based participatory approaches could be applied in NCD and wider health research in Sub-Saharan Africa but also the strengths of co-creation in art-based participatory research as a new way to generate new rich, in-depth understandings.

However, there are some limitations. First, as arts-based research can encompass a diverse range of activities, the findings may not be generalisable of the findings beyond the activities employed in the current study. Second, as both communities have had long involvement in health research, participants may have had increased exposure to health information compared to the general populations of Malawi and Tanzania. Although people's knowledges, attitudes and behaviours will undoubtably be influenced by prior experience of health research, there is no reason to believe that the arts-based participatory approach described here should not be translatable to other communities with different levels of pre-existing health knowledge. Third, there were some differences in the questions explored in the workshops and focus group discussions. However, it is unlikely that these differences affected the conclusions about the emergence of more in-depth content and the reduction of participant-researcher hierarchies within the workshops. Finally, the lead author (MB) was not present during the workshops or focus groups and thus may have missed important contextual cues, particularly during the arts-based participatory activities. However, the analysis was closely supported by the lead Tanzanian researcher (SMM) who was present at both Tanzanian workshops and informed by fieldnotes describing researcher experiences and perceptions of data collection. The lead Malawian researcher (OM) was also consulted closely to ensure adequate representation of local context in the findings presented.

## Conclusions

Compared to traditional focus group discussions, the arts-based participatory workshops revealed more emotive and personal narratives of these NCD experiences. They appeared to liberate participants from the trappings of cognitive ways of knowing and generate new in-depth understandings. The workshops also succeeded in empowering participants to voice their own perspectives, including fears and negative beliefs, and in supporting the development of a performance that reflected participants' personal interests and concerns about NCDs. As such, the arts-based participatory workshops show promise as a new approach to inform the development of culturally relevant NCD prevention initiatives among at-risk populations in Sub-Saharan Africa and potentially elsewhere.

## Supporting information

**S1 Checklist. Standards for Reporting Qualitative Research (SRQR) checklist.**
(DOCX)

**S1 Table. Full descriptions of the nine themes identified during the thematic analysis of the arts-based participatory workshop and focus group transcripts.**
(DOCX)

## Acknowledgments

We would like to thank Hazel Namadingo from MEIRU for her contributions to data collection, Bosco Chikonda from ArtGlo for his operational support and Jo Sharp from the University of St Andrews and Chisomo Kalinga from University of Edinburgh for their conceptualisation of the project.

## Author Contributions

**Conceptualization:** Cindy M. Gray, Sharifa Abdulla, Christopher Bunn, Amelia C. Crampin, Angel Dillip, Jason M. R. Gill, Elson Kambalu, John Lwanda, Mia Perry, Zoë Strachan, Helen Todd, Sally M. Mtenga.

**Data curation:** Angel Dillip, Otiyela Mtema, Sally M. Mtenga.

**Formal analysis:** Maria Bissett, Cindy M. Gray, Sharifa Abdulla, Christopher Bunn, Amelia C. Crampin, Angel Dillip, Jason M. R. Gill, Sharon Kalima, John Lwanda, Otiyela Mtema, Mia Perry, Zoë Strachan, Helen Todd, Sally M. Mtenga.

**Funding acquisition:** Cindy M. Gray, Sharifa Abdulla, Christopher Bunn, Amelia C. Crampin, Angel Dillip, Jason M. R. Gill, Elson Kambalu, John Lwanda, Mia Perry, Zoë Strachan, Helen Todd, Sally M. Mtenga.

**Investigation:** Cindy M. Gray, Angel Dillip, Heri C. Kaare, Sharon Kalima, Herbert F. Makoye, Otiyela Mtema, Sally M. Mtenga.

**Methodology:** Cindy M. Gray, Sharifa Abdulla, Christopher Bunn, Amelia C. Crampin, Angel Dillip, Elson Kambalu, John Lwanda, Otiyela Mtema, Mia Perry, Zoë Strachan, Helen Todd, Sally M. Mtenga.

**Supervision:** Cindy M. Gray, Sally M. Mtenga.

**Validation:** Cindy M. Gray, Sally M. Mtenga.

**Writing – original draft:** Maria Bissett, Cindy M. Gray, Sally M. Mtenga.

**Writing – review & editing:** Sharifa Abdulla, Christopher Bunn, Amelia C. Crampin, Angel Dillip, Jason M. R. Gill, Heri C. Kaare, Sharon Kalima, Elson Kambalu, John Lwanda, Herbert F. Makoye, Otiyela Mtema, Mia Perry, Zoë Strachan, Helen Todd.

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
