## [Decision Letter · Decision Letter 0]

29 Sep 2022

*"I see salt everywhere"*: A qualitative examination of the utility of arts-based participatory workshops to study noncommunicable diseases in Tanzania and Malawi.

PGPH-D-22-01204

Dear Dr. Gray,

We are pleased to inform you that your manuscript '*"I see salt everywhere"*: A qualitative examination of the utility of arts-based participatory workshops to study noncommunicable diseases in Tanzania and Malawi.' has been provisionally accepted for publication in PLOS Global Public Health.

Best regards,

Maurizio Trevisan, M.D., MS

Academic Editor

Reviewer Comments (if any, and for reference):

Reviewer's Responses to Questions

**Comments to the Author**

1. Does this manuscript meet PLOS Global Public Health’s publication criteria? Is the manuscript technically sound, and do the data support the conclusions? The manuscript must describe methodologically and ethically rigorous research with conclusions that are appropriately drawn based on the data presented.

Reviewer #1: Yes

2. Has the statistical analysis been performed appropriately and rigorously?

Reviewer #1: N/A

3. Have the authors made all data underlying the findings in their manuscript fully available (please refer to the Data Availability Statement at the start of the manuscript PDF file)?

Reviewer #1: Yes

4. Is the manuscript presented in an intelligible fashion and written in standard English?

Reviewer #1: Yes

5. Review Comments to the Author

Reviewer #1: A Novel approach towards thorough understanding of participants’ beliefs and emotions about NCDs in this study. Although the results or findings obtained through this study cannot be applied because of the limitations clearly cited in this study, the approach/method used in the study can be further explored to generate better understanding of stigmas associated with health-related issues in different cultures. The study can be taken as a baby step towards generating a multidisciplinary approach for the betterment of a multicultural society. Further studies conducted in similar lines with diverse groups could come up with better solution-oriented ideas. For that, it’s better to list out problems like stigma and false beliefs associated with health among a population, and then conduct such studies for a better understanding and clarity.

6. PLOS authors have the option to publish the peer review history of their article (what does this mean?). If published, this will include your full peer review and any attached files.

**Do you want your identity to be public for this peer review?** For information about this choice, including consent withdrawal, please see our Privacy Policy.

Reviewer #1: **Yes: **Dr. Madhuri Devaraju
